# (Ab)use of Health Claims in Websites: The Case of Italian Bottled Waters

**DOI:** 10.3390/ijerph16173077

**Published:** 2019-08-24

**Authors:** Giulia Lorenzoni, Clara Minto, Matteo Temporin, Elisa Fuscà, Anna Bolzon, Gianluca Piras, Sabino Iliceto, Marco Silano, Dario Gregori

**Affiliations:** 1Unit of Biostatistics, Epidemiology and Public Health, Department of Cardiac, Thoracic, Vascular Sciences and Public Health, University of Padova, 35131 Padova, Italy; 2Unit of Cardiology, Department of Cardiac, Thoracic, Vascular Sciences and Public Health, University of Padova, 35128 Padova, Italy; 3Italian National Institute of Health, 00161 Rome, Italy

**Keywords:** website content, bottled water, health claims, nutrition claims, consumers

## Abstract

The massive use of web marketing makes the monitoring of nutrition and health claims used in advertising campaigns much more difficult. The present study aimed at reviewing the website content for bottled waters produced in Italy to assess (i) if nutrition and health claims are reported, (ii) what types of nutrition and health claims are reported most frequently, and (iii) if the nutrition and health claims could be considered appropriate according to the current regulation in the field. A review of the website content of the 253 bottled waters produced in Italy and reported in the annual report of Bevitalia 2016–2017 was conducted. For each brand, indications related to the preventive, curative or therapeutic properties of the water reported were examined. Bottled waters that included potentially misleading information apparently not consistent with the European Directive on the exploitation and marketing of natural mineral waters were identified. Forty bottled waters with uncertain website content were identified. The information reported in the websites referred most often to beneficial effects for urinary tract and cardiovascular systems. Present results highlight, using the bottled water case study, that website content sometimes happens to deliver misleading information to consumers, also thanks to uncertain regulation in this sensitive field.

## 1. Introduction

Consumption and demand for bottled water have grown exponentially in the last few years. In 2016, the U.S. market recorded an increase of nearly 9% in total bottled water volume, which grew from 44.6 billion liters in 2015 to 47.6 billion liters in 2016 [1]. In the same year, Europe reconfirmed its leadership position in the consumption and production of bottled water, with about 109.9 liters per capita yearly consumption, and 594 small and medium producers [2]. Among European countries, Italy has the highest number of natural mineral waters, with more than 250 different brands, and an estimated per capita yearly consumption of 188.5 liters [3,4]. 

Reasons for this growing trend are still uncertain. Some authors highlighted how consumer preferences could be influenced by perceived safety, purity, and taste of bottled water [5,6]. Marketing strategies also seem to have played a role in enhancing the consumption trends of bottled waters. Each year, bottled water companies spend billions of dollars to create appealing advertisements for different communication channels such as print, television, and web advertising [5]. Among such “appealing advertisings” there are also nutrition and health claims which have become new marketing tools [7] whose effects on consumers’ choices are still under study [8,9,10]. 

Given the widespread use of nutrition and health claims, the European Commission has legislated the use of such claims to ensure consumers’ protection (Regulation 2006/1924/EC [11] and Directive 2009/54/EC [12]; the latter adopted by the Italian government with the D. Leg. N. 176 of 2011 [13]). The Directive 2009/54/EC [12] states that indications relating the mineral water to the prevention, treatment, or cure of human diseases are prohibited, and only a limited number of indications could be authorized according to strict criteria. After 2009, bottled water producers progressively adapted water labels to the new EU requirements. However, in the same period, marketing strategies have undergone dramatic changes represented by the increasing use of web advertising along with traditional strategies (e.g., TV advertising) [14]. The massive use of web marketing makes the monitoring of advertising campaigns much more difficult. The monitoring of website content is still challenging [15] due to its flexibility and the easiness of implementing modifications in its content, potentially reducing consumers’ protection against misleading information. 

The use of misleading health and nutrition claims represents an emerging public health issue, especially concerning the use of misleading information in the digital media, and the literature in the field is scant. Even though regulatory bodies in both the U.S. and Europe allow the use of only scientifically sound health and nutrition claims, it has been recently claimed that the online distribution is not adequately monitored in the European Union. This seems to be testified by a high prevalence of misleading information posted on the websites that could have relevant consequences for the consumers, also in terms of health effects of marketed products [16]. For these reasons, the need for stricter surveillance of the claims used by companies to promote their products has been advocated [17]. Unfortunately, the use of false health and nutrition claims to promote food and beverages is not uncommon, and those most commonly involved seem to be breakfast products, e.g., ready-to-eat cereals and fruit juices [18]. Recently, in Italy, the Italian Competition Authority (ICA) financially sanctioned the use of misleading claims and advertising of seven different products, mainly biscuits and yogurts [19]. Not least, it has been shown that the use of health messages to advertise unhealthy food products is growing, especially aimed to children [20]. Worryingly, the use of such claims seems to affect consumers’ choices significantly; a recent study has shown that the demand for four products that underwent a Federal Trade Commission (FTC) investigation for the use of false claims declined dramatically after the brand stopped the use of the claim under investigation [21]. 

Considering such a framework, it becomes clear the need for a better understanding of the extent of the use of misleading health and nutrition claims to promote food and beverages. This work presents a case study aimed at reviewing the website contents of bottled waters produced in Italy to assess (i) if nutrition and health claims are reported, (ii) what types of nutrition and health claims are reported most frequently, (iii) if the nutrition and health claims could be considered appropriate with respect to the evidence needed according to the current regulation in the field. 

## 2. Materials and Methods 

A review of the websites of the 253 bottled waters reported in the annual report of Bevitalia 2016–2017 [4] was performed. The Bevitalia report provides detailed information about the Italian market for soft drinks and bottled waters (e.g., products, manufacturers, consumption).

The review was performed between October 2016 and January 2018. Bottled water websites that included potentially misleading information apparently not consistent with European Directive on the exploitation and marketing of natural mineral waters [12] were identified. 

### 2.1. Coding Procedures

The website contents of bottled waters were considered to be consistent with the EU Directive 2009/54/EC [12] if they reported a health indication similar to: May be laxativeMay be diureticMay facilitate the hepato-biliary functionsMay stimulate digestion

These statements represented the authorized indications. A very conservative approach was chosen, deciding to consistently consider the vague presence of any of the above statements, despite the regulatory mandate for which such claims should be used only on the basis of stricter criteria. 

On the other side, website contents of the bottled waters were considered to be not consistent (i.e., “uncertain”) to the EU Directive if they reported health claims related to: PreventiveCurativeTherapeutic properties

These claims are not allowed by the legislation. Particularly, all information with uncertain wording was included, such as ability to (“capacità di” in Italian language), treatment of (“cura di”), effective for (“efficace per”), suitable for (“adatto a”), perfect for (“ideale per”), useful (“utile”), contributes to (“constribuisce a”), facilitates (“favorisce”, “stimola”), prevents (“previene”), allows (“consente”, “permette”), acts (“agisce”), and improves (“migliora”). Additionally, for bottled waters with uncertain website content, a review of references to scientific evidence, pharmacological examinations, and clinical trials supporting reported health claims was performed.

The extraction of the information from the websites and the evaluation of the compliance with the legislation was performed in double (A.B. and E.F). If any disagreement occurred, it was solved by a third independent reviewer (G.P.). The evaluation of the compliance with the legislation was performed according to a checklist developed for the purpose of the study. The checklist was developed based on the criteria mentioned above. According to the checklist criteria, the website content could be coded as “consistent to the legislation” or “uncertain”. 

### 2.2. Statistical Methods

The proportion of brands with uncertain website content was reported along with 95% Confidence Interval (C.I.) [22]. The agreement between the reviewers was evaluated using the Cohen’s Kappa [23]. To evaluate the association between sale prices and probability of “uncertain” claims, a logistic regression model has been fitted. Results are presented as Odds Ratio (OR) with 95% C.I. All analyses have been made with the R-System (R Foundation for Statistical Computing, Vienna, Austria) [24].

## 3. Results

Among the 253 brands reviewed (reported in the report of Bevitalia 2016–2017), 40 bottled waters (15.8%, 95% C.I. 11.5%–21%) with uncertain website content were identified. For the sake of clarity and transparency, the database reporting the identified bottled waters with uncertain website content is available at: https://1drv.ms/u/s!Au30_jLUzf-rp-0SnlYOH3nJjEGXGQ?e=q6IJkv. The agreement between the reviewers was excellent (Cohen’s Kappa = 0.91).

Thirty-six brands included normal mineral water; the others included thermal waters for mineral water treatment. Information identified as “uncertain” was associated with a description of a relationship between water consumption and specific biochemical parameters or of regulatory mechanisms of the human body and pathological conditions (Table 1). 

.

Prices were highly heterogeneous among waters, ranging from 0.07€ up to 7€ per liter, with prices being higher for waters with “uncertain” claims (median per liter 0.48€ vs. 0.38€ for waters with “consistent to the legislation” claims). The probability of using “uncertain” claims was not significantly associated with sale prices (*p* = 0.175), although a positive trend (the higher the price the higher the probability of such claims) can be observed (Figure 1).

Biochemical parameters included hematological and urinary values such as uric acid, free radicals, nitrogen level, creatinine, cholecystokinin, calcium, lactic acid, blood viscosity, blood oxides and general metabolic toxins. Waters contributing to modulation of regulatory mechanisms reported to have effects on generic metabolic functions, the ageing process, biliary secretion or immuno-inflammatory cascade. Table 2 shows the distribution of the claimed beneficial effects reported in the websites, showing that information reported in the websites referred most often to beneficial effects for urinary tract and cardiovascular systems. 

Moreover, in most cases, the website claimed that the properties of water are certified by the Italian Ministry of Health, by university researchers, or by international studies. Nonetheless, in a very few cases we succeeded to find those studies, and only in two cases, the website offers the full documentation signed by the Italian Ministry of Health.

Finally, many websites have an Italian version and an English one. The check of the translation revealed that often the information written in Italian was not the same as reported in the international version of the website. The English version is often enriched with descriptions, characteristics, and statistics whose scientific origin is most often not provided.

## 4. Discussion

The present study aimed at understanding the claimed health benefits of bottled water as presented in the corresponding products’ websites, to urge a discussion on the implications for communicating food-related health effects on the Internet. This was accomplished by an exhaustive review of the website content of the bottled waters produced in Italy. Present results showed that about one sixth of brand’s websites reported information classified as “uncertain”, meaning by that a borderline adherence to the EU directive on health claims for bottled water. Such “uncertain” beneficial effects referred most often to the urinary tract and cardiovascular system, indicating, in terms of plausibility, an association among the chemical characteristics of drinking water and some biological functions, especially the cardiovascular, digestive, and urinary ones. Evidence in this field is, however, still debated and should be taken with caution [25]. This is the reason behind the introduction of strict criteria to which industry should conform, and the consequent small set of indications which have been authorized for bottled waters [12]. According to the variety of health effects (indicated in Table 2) ranging from generic “suggestions” to strong statements, and the diversity of biological mechanisms indicated, there is no doubt that website marketing represents an empty territory in which regulation should be made more effective and perhaps even re-conceptualized. Indeed, the influence of website contents on consumers’ choices and behaviors is a relatively new topic of growing interest and still under study [26,27,28]. It has been shown that the marketing of food and beverages through the new digital media is going to play an increasingly important role, especially among young people [29]. This highlights the need for monitoring website content strictly, since, as it has been shown in the present study, they could be misleading. 

In Italy, consumer protection against misleading advertising and fraudulent commercial practices is entrusted by the Antitrust and Market Authority (AGCM), a public authority, and the private Institute of Self-Discipline Advertising (IAP). Recently, the IAP has raised four injunctions to water companies for the deceptive content that did not comply with the advertising self-discipline code. These facts show how traditional marketing strategies (labels and TV advertising) are already under strict surveillance by the regulatory authorities. The same attention should also be placed on website content, given the growing efforts made by companies in implementing new channels of advertising using digital media, and the growing interest of consumers to improve their knowledge about the characteristics of the products they are going to purchase. This is crucial to enhance the consumer’s protection from misleading and incorrect informative practices.

It is worth pointing out that the present work is based on a case study about misleading claims reported on the websites of Italian bottled waters. This poses relevant limitations for the generalizability of the results outside Italy and with regards to other types of food and beverage products. However, such aspects fall outside the study’s aim, which was to present a case study that serves as an example of an emerging public health issue, that of misleading website content. Not least, a very conservative approach has been adopted in coding the website content as “consistent to the legislation” or “uncertain”, so we cannot rule out that the phenomenon of misleading claims reported in the websites of Italian bottled waters has been underestimated. 

## 5. Conclusions

The results of the present study highlight, using the bottled water case study, that website content sometimes delivers misleading information to consumers, also thanks to an uncertain regulation in this sensitive field. 

The use of misleading health and nutrition claims on websites is an emerging public health issue that requires the attention of legislative authorities. This might translate into the need for (i) developing surveillance systems for the detection of misleading claims on websites, (ii) developing an ad-hoc regulation toward approval of web-content related to health claims of food and, for instance, of bottled waters, (iii) establishing sanctions for manufacturers that abuse health and nutrition claims in websites.

Further research should focus on investigating website content of other product categories also outside Italy and on better characterizing the impact of the use of misleading claims on websites on consumers’ choices. 

## Figures and Tables

**Figure 1 ijerph-16-03077-f001:**
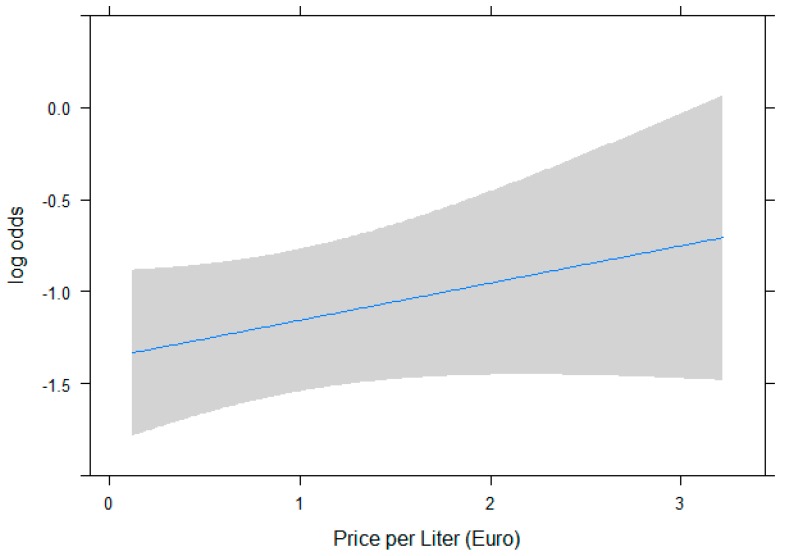
Association between price of water and risk of “uncertain” claim. Association is not statistically significant (OR for a 0.5€ increase is 1.10, 95% C.I. 0.955–1.280).

**Table 1 ijerph-16-03077-t001:** Wording included in website of the selected bottled water (*n* = 40).

Water Brand	Web Site	Web Site Content
Acqua Filette	www.acquafilette.it	Reduces blood viscosity. Improves metabolic functions.
Acqua Leo	www.acqua-leo.it	Strongly suggested for heart failure, hypertension, hepatic cirrhosis.
Acqua Maxim’s	www.acquamaxims.com	Really effective for cardiovascular problems and hypertension. Helps to expel of blood oxides.
Acqua Santa Chianciano (mineral water treatment)	www.termechianciano.it	Contributes to the elimination of long-term effects of pharmacological therapies.
Amata	www.acquaamata.it	Contributes to the reduction of CVD risk Suggested for hypertension
Amorosa	www.acquafonteviva.it	Prevents diseases related to oxidation processes, diseases related to alkalizing processes and diseases related to toxins accumulation.
Azzurrina	www.fonteazzurrina.com	Allows the reduction of kidney diseases. Has the ability to rebalance the acidity of organism. It is the best water for subjects with celiac disease. Effective against free radicals. It slows down skin ageing process. Prevents diseases such as diabetes, arthritis, hypertension and neoplasm.
Bracca	www.fontebracca.it	Able in promoting the elimination of uric acid and digestive processes in general
Castellina	www.castellina.it	Facilitates the elimination of uric acid. Prevents kidney stones. Facilitates the emptying of gallbladder. Suggested for calculus of gallbladder
Castello	www.acquacastello.it	Prevents the formation of salt crystals in kidneys. Enables prevention and assistance in the treatment of kidney stones and renal gravel.
Cerelia	www.acquacerelia.com	Is medically recommended for pathological diseases of the urinary tract (lithiasis, cytis, pyelitis) and of the liver.
Fiuggi	www.acquadifiuggi.eu	Can act on renal and articular consequences of hyperuricemia. Can be useful for kidney stones. It can be supposed that the water stimulates the dilatation of ureter, thus facilitating an easier expulsion of small kidney stones.
Fonte Essenziale	www.fonteessenziale.it	Suggested for subjects with non-pathological digestive disorders, constipation irritable bowel, diverticular disease, gallbladder disorders. Facilitates the secretion of Cholecystokinin. Facilitates the action of gastrin.
Fonte Noce	www.fontenoce.it	Clinically indicated for kidney stones, chronic inflammation of urinary tract, glomerulonephritis, clinical manifestations of gout, uric acid diathesis, oxalic diathesis.
Fonte Ventasso	www.grupposem.it	Has anti-inflammatory and anti-toxic effect.
Fonteviva	www.acquafonteviva.it	Prevents kidney stones. Acts against oxidation process and cellular ageing process. Helps to eliminate toxins.
Fucoli (mineral water treatment)	www.termechianciano.it	Has anti-inflammatory action on the gastroduodenal mucosa. Contributes to prevention of osteoporosis.
Gaverina Fonte Centrale	www.fontidigaverina.it	Has anti-inflammatory and antispasmodic action. Effective for digestive disorders, dyspepsia, gastritis, enterocolitis, spastic colitis, constipation, diarrhea.
Kaiserwasser	www.kaiserwasser.com	Suggested for subjects with digestive disorders. Suggested for subjects who suffer of meteorism and for body weight control. Is more effective if taken during meals. Useful for treatment of constipation.
Lauretana	www.lauretana.com	Helps organism to eliminate uric acid and creatinine. Helps to prevent the formation of kidney stones. Contributes to the prevention of oxidation processes. Suggested for subjects with hypertension, heart failure, kidney failure, cellulite and water retention.
Lete	www.acqualete.it	Suggested for subjects with hypertension and osteoporosis.
Levico	www.levicoacque.it	Suggested for subjects with hypertension.
Mangiatorella	www.mangiatorella.it	Suggested for prevention of kidney stones and other kidney diseases. Contributes to slowing down ageing process of articulations, muscles, skin and hair. Mangiatorella is rich of silica mineral, a substance that can reduce risk of dementia and Alzheimer.
Monte Cimone	www.grupposem.it	Useful to prevent hypertension. Has indirect anti-inflammatory action and anti-toxic effect.
Orobica	www.fontiprealpi.com	Can facilitate the elimination of uric acid.
Pejo	www.pejo.it	Perfect for kidney stones.
Pioda	www.acquamineralestellalpina.com	Suggested for subjects with hypertension or obesity, and for subjects with heartburn.
Roana	www.acquaroana.it	Suggested for treatment of hypertension. Perfect for treatment of kidney stones. Facilitates calcium “fixation”. Reduces blood pressure.
Rocchetta Naturale	www.rocchetta.it	Clean the body from metabolic slags. Helps keep the skin youth and beautiful. Gives a bright look.
S. Lucia	www.acquasantalucia.it	Has vital energy.
Sancarlo Spinone	www.spumador.com	Suggested for remineralization of tissues and bones
Sangemini	acquemineraliditalia.it	Helps maintain bone heritage (reducing the risk of osteoporosis and fracture) Improves muscle contraction and neuromuscular excitability Promotes the activation of hormones, such as insulin Helps regulate blood pressure Helps to coagulate the blood Contributes to slowing down ageing process, and to strengthen the hair.
Santa Croce (mineral water treatment)	www.acquasantacroce.it	Facilitates the elimination of uric acid. Suggested for recurrent cystitis, chronic inflammation of the urinary tract, urinary lithiasis, biliary dyspepsia, biliary microlithiasis, gallbladder, inflammation of biliary tracts.
Santa Maria	www.acquasantamaria.it	Can positively act on dyspepsia.
Suio	www.suio.it	Important action against muscle cramps. Has positive effects against atherosclerosis and osteoporosis.
Surgiva	www.surgiva.it	Suggested for prevention of osteoporosis and hypertension.
Tavina	www.tavina.it	Improves clinical conditions of patients with dyspepsia or digestive disorders.
Telese (mineral water treatment)	www.termeditelese.it	Useful for prevention of osteoporosis. Suggested in case of gastritis, hepatic disorders and constipation.
Tesorino	www.tesorino.com	Effective in treatment of kidney diseases and digestive disorders. Suggested for hypercholesterolemia, gastritis, peptic ulcer, kidney stones.
Tolentino Sorg. S. Lucia (mineral water treatment)	www.termesantalucia.it	Suggested for kidney diseases, kidney stones, and clinical manifestations of gout. Has the ability to dilute the urine. Increases motility of ureters and facilitate the elimination of small kidney stones.

**Table 2 ijerph-16-03077-t002:** Wording included in website of the selected bottled water (*n* = 40).

Beneficial Effects Advertised on Websites		Number of Brands
**Prevention of**	Kidney/urinary tract diseases	5
	Oxidation processes	2
	Diabetes, arthritis, neoplasm	1
	Osteoporosis	3
	Hypertension	3
**Suggested for**	Hypertension	7
	Kidney/urinary tract diseases, gout	6
	Digestive disorders/gastrointestinal diseases	5
	Gallbladder diseases	3
	Hypercholesterolemia	1
	Hepatic diseases	3
	CV diseases	2
	Obesity/body weight control	2
	Osteoporosis	2
**Reduction of risk/incidence of**	CV diseases	1
	Kidney/urinary tract diseases	1
	Dementia and Alzheimer	1
	Osteoporosis	1
**Effective/useful/ perfect for**	CV diseases	1
	Hypertension	1
	Celiac disease	1
	Digestive disorders/gastrointestinal diseases	4
	Kidney/urinary tract diseases	5
	Osteoporosis	2
	Atherosclerosis	2
**Other actions**	Improves metabolic functions	1
	Contributes to eliminate long-term effects of drugs	1
	Rebalances the acidity of organism	1
	Slows down aging process	2
	Facilitates emptying of gallbladder	2
	Facilitates elimination of uric acid	4
	Facilitates biliary secretion	1
	Facilitates action of gastrin	1
	Action against oxidation process	1
	Anti-inflammatory action	4
	Antispasmodic action	1
	Antitoxic action	3
	Helps keep the skin youth and beautiful	1
	Has vital energy	1
	Improves muscle contraction and neuromuscular excitability	1
	Promotes the activation of hormones, such as insulin	1
	Helps regulate blood pressure	1
	Helps to coagulate the blood	1
	Contributes to slowing down ageing process, and to strengthen the hair	1
	Positively acts on dyspepsia	1
	Action against muscle cramps	1
	Improves clinical conditions of patients with digestive disorders	1
	Facilitates elimination of kidney stones	1

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
