# Peer review of "(Ab)use of Health Claims in Websites: The Case of Italian Bottled Waters"

_ijerph, 2019, doi:10.3390/ijerph16173077_

Round 1
Reviewer 1 Report
I thank the authors for tackling an important issue. Based on my reading of the paper, I have provided my suggestions for improvement.
The article is missing a literature review. As is, I see the introduction followed by the method. A strong literature summarizing past and relevant studies should be provided after the introduction.
In the literature, explain the importance of studying misleading claims, esp., in Italy.
I am concerned about the method. Now, it would seem a content analysis was done, however, procedural information like the codebook, number of coders, intercoder reliability, etc. are not mentioned. The rigor of this section requires to be strengthened, else, the results don't hold value.
The authors mention looking at website content but in the same sentence they also add 'online advertisements,' which is incorrect. The unit of analysis is the information on the website and not the online advertisements of the brands. The distinction has to be clear regarding what was and what was 'not' coded.
In the results section, it would seem the findings were descriptive with simple frequencies reported. This is important, however, I invite the authors to revisit the data and see if running Chi-square analysis would help add depth to the findings.
With changes to the literature review and the additional analysis ('Chi-square'), the discussion section too can be significantly strengthened.
Limitations and future research are missing. Please provide the same.
I wish the authors the very best.
Author Response
The article is missing a literature review. As is, I see the introduction followed by the method. A strong literature summarizing past and relevant studies should be provided after the introduction. In the literature, explain the importance of studying misleading claims, esp., in Italy.
The introduction has been revised by including relevant studies in the field.
I am concerned about the method. Now, it would seem a content analysis was done, however, procedural information like the codebook, number of coders, intercoder reliability, etc. are not mentioned. The rigor of this section requires to be strengthened, else, the results don't hold value.
A description of the coding procedures has been provided in the Materials and Methods section
We would like to thank the reviewer for the comment. The issue was an incorrect wording using the words “online advertising”. This language mistake ended up in a little confusion. Actually the unique unit of the analysis was the website content. The manuscript has been amended accordingly.
We thank the reviewer for the helpful comment. Indeed we added some inferential information to help the reader to broaden the context of the study. In this sense, 95% Confidence Interval was provided to improve results generalizability. However, it is worth pointing out that the present work is based on a case study about misleading claims reported in the websites of Italian bottled waters. This poses relevant limitations for the generalizability of the results outside Italy and with regards to other type of food and beverage products. However, such aspect falls outside the study’s aim, which was to present a case study that serves as an example of an emerging public health issue. Such information has been discussed in the section.
Study limitations have been provided in the Discussion section. Suggestions for future research have been provided in the Conclusions.
Reviewer 2 Report
The authors have conducted a content analysis of the online advertising of bottled water produced in Italy. The biggest concern for me is the value of this paper. The authors should highlight the value of this study. The authors can add implications for the regulator and also the marketer. Furthermore, I have some suggestions for further revisions:
Please give a brief description of Bevitalia 2016-2017.
The method session is very not detailed enough. "indications related to preventive, curative or therapeutic properties of water reported in official websites were examined." The authors can give some examples of preventive, curative and therapeutic properties. How many coders were involved in this process? Is there a test conducted for inter-coder reliability? The criteria for coding should be more specific.
Please provide implications for regulators, legislators, or marketers. Simply saying "the EU legislators should consider an ad-hoc regulation toward approval of web-content related to health claims of food...." is too general.
This paper should refer to some relevant papers from IJERPH or relevant journals.
Author Response
Please give a brief description of Bevitalia 2016-2017.A brief description of the report has been provided in the Materials and Methods section.
A more detailed description of the coding procedures has been provided in the Materials and Methods section
A more detailed descriptions of the most important implications of the present study has been provided in the Conclusion section.
We included some references as recommended by the reviewer briefly discussing them.
Round 2
Reviewer 1 Report
Thank you for making the requested changes. The importance and quality has been enhanced, however, I do have a few more comments:
1) In the newly added lit. review, the sources are primarily based from the US with reference to the FTC and FDA, which is ok. However, since the country of focus is Italy, regulations can be different. Are there similar studies done in the EU that can be used instead?
2) The studies chosen are focused on information from ads and packaging, however, the study's focus is on website information, which is different. Again, are there articles that have studied brand websites (vs. ads and food packaging)? This makes the literature much tighter.
3) In the method, please specify how many coders or reviewers were involved in the coding.
4) The discussion delves more deeply into online advertising, which is again an incorrect reference to website information. I would ask the authors to re-read the paper and make corrections accordingly.
5) Considering that IJERPH is a top journal, I still struggle with the simple report of frequencies. I am not undermining your work, the results do hold value and I understand that the objective was to focus only on website content. In the future, I would urge the authors to explore the current dataset or conduct follow-up studies with a more comprehensive codebook that looks at other categories, e.g., a) dividing the bottled water brands into luxury [Evian or FIJI] vs. regular [Dasani]. Are all bottled brands providing deceptive information or is it the top brands? What does the Chi-Square analysis show? b) You could even categorize the medical claims themselves and see which creates the greatest impact.
Wish the author(s) the very best!
Author Response
1) In the newly added lit. review, the sources are primarily based from the US with reference to the FTC and FDA, which is ok. However, since the country of focus is Italy, regulations can be different. Are there similar studies done in the EU that can be used instead?
Done.
2) The studies chosen are focused on information from ads and packaging, however, the study's focus is on website information, which is different. Again, are there articles that have studied brand websites (vs. ads and food packaging)? This makes the literature much tighter.
Done
3) In the method, please specify how many coders or reviewers were involved in the coding.
Three coders were involved. Two were involved in the coding procedures and the third one was involved in the consensus agreement if disagreements occurred. Such information is reported in the Materials and Methods section: “The extraction of the information from the websites and the evaluation of the compliance with the legislation were performed in double (A.B. and E.F). If any disagreement occurred, it was solved by a third independent reviewer (G.P.).”
4) The discussion delves more deeply into online advertising, which is again an incorrect reference to website information. I would ask the authors to re-read the paper and make corrections accordingly.
Done
5) Considering that IJERPH is a top journal, I still struggle with the simple report of frequencies. I am not undermining your work, the results do hold value and I understand that the objective was to focus only on website content. In the future, I would urge the authors to explore the current dataset or conduct follow-up studies with a more comprehensive codebook that looks at other categories, e.g., a) dividing the bottled water brands into luxury [Evian or FIJI] vs. regular [Dasani]. Are all bottled brands providing deceptive information or is it the top brands? What does the Chi-Square analysis show? b) You could even categorize the medical claims themselves and see which creates the greatest impact.
To improve results generalizability, it has been performed an evaluation of the association between sale prices and probability of “uncertain” claims, a logistic regression model has been fitted. Prices were highly heterogeneous among waters, ranging from 0.07€ up to 7€ per liter) being higher for water with “uncertain” claims, median (per liter) 0.48€ vs. 0.38€ for waters with “consistent to the legislation” claims. The probability of using “uncertain” claims was not significantly associated with sale prices (p=0.175), although a positive trend (the higher the price the higher the probability of such claims) can be observed Such information has been added in the manuscript.
Reviewer 2 Report
The quality of the paper has been improved. The authors have addressed all the problems. Of course, the paper can still be improved by adding more relevant literature in the introduction section.
Author Response
The quality of the paper has been improved. The authors have addressed all the problems. Of course, the paper can still be improved by adding more relevant literature in the introduction section.
Done